# Automatic Integration for Spatiotemporal Neural Point Processes

**Zihao Zhou**
Department of Computer Science
University of California, San Diego
La Jolla, CA 92092
ziz244@ucsd.edu

**Rose Yu**
Department of Computer Science
University of California, San Diego
La Jolla, CA 92092
roseyu@ucsd.edu

## Abstract

Learning continuous-time point processes is essential to many discrete event fore-casting tasks. However, integration poses a major challenge, particularly for spatiotemporal point processes (STPPs), as it involves calculating the likelihood through triple integrals over space and time. Existing methods for integrating STPP either assume a parametric form of the intensity function, which lacks flexibility; or approximating the intensity with Monte Carlo sampling, which in-troduces numerical errors. Recent work by Omi et al. [2019] proposes a dual network approach for efficient integration of flexible intensity function. However, their method only focuses on the 1D temporal point process. In this paper, we introduce a novel paradigm: `AutoSTPP` (Automatic Integration for Spatiotempo-ral Neural Point Processes) that extends the dual network approach to 3D STPP. While previous work provides a foundation, its direct extension overly restricts the intensity function and leads to computational challenges. In response, we introduce a decomposable parametrization for the integral network using ProdNet. This approach, leveraging the product of simplified univariate graphs, effectively sidesteps the computational complexities inherent in multivariate computational graphs. We prove the consistency of `AutoSTPP` and validate it on synthetic data and benchmark real-world datasets. `AutoSTPP` shows a significant advantage in recovering complex intensity functions from irregular spatiotemporal events, particularly when the intensity is sharply localized. Our code is open-source at https://github.com/Rose-STL-Lab/AutoSTPP.

## 1 Introduction

Spatiotemporal point process (STPP) [Daley and Vere-Jones, 2007, Reinhart, 2018] is a continuous time stochastic process for modeling irregularly sampled events over space and time. STPPs are par-ticularly well-suited for modeling epidemic outbreaks, ride-sharing trips, and earthquake occurrences. A central concept in STPP is the *intensity function*, which captures the expected rates of events occur-rence. Specifically, given the event sequence over space and time $\mathcal{H}_t = \{(\mathbf{s}_1, t_1), \ldots, (\mathbf{s}_n, t_n)\}_{t_n \leq t}$, the joint log-likelihood of the observed events is:

$$\log p(\mathcal{H}_t) = \sum_{i=1}^{n} \log \lambda^*(\mathbf{s}_i, t_i) - \int_{\mathcal{S}} \int_0^t \lambda^*(\mathbf{u}, \tau) d\mathbf{u} d\tau \qquad (1)$$

$\lambda^*$ is the optimal intensity, $\mathcal{S}$ the spatial domain, $\mathbf{u}$ and $\tau$ the space and time, and $t$ the time range.

Learning STPP requires multivariate integrals of the intensity function, which is numerically challeng-ing. Traditional methods often assume a parametric form of the intensity function, such as the integral

can have a closed-form solution Daley and Vere-Jones [2007]. But this also limits the expressive power of the model in describing complex spatiotemporal patterns.

Others propose to parameterize the model using neural ODE models Chen et al. [2020] and Monte Carlo sampling, but their computation is costly for high-dimensional functions. Recently, work by Zhou et al. [2022] proposes a nonparametric STPP approach. They use kernel density estimation for the intensity and model the parameters of the kernels with a deep generative model. Still, their method heavily depends on the number of background data points chosen as a hyper-parameter. Too few background points cause the intensity function to be an inflexible Gaussian mixture, while too many background points may cause overfitting on event arrival.

To reduce computational cost while maintaining expressive power, we propose a novel *automatic integration* scheme based on dual networks to learn STPPs efficiently. Our framework models intensity function as the sum of background and influence functions. Instead of relying on closed-form integration or Monte Carlo sampling, we directly approximate the integral of the influence function with a deep neural network (DNN). Taking the partial derivative of a DNN results in a new computational graph that shares the same parameters; see Figure 1.

First, we construct an integral network whose derivative is the intensity function. Then, we train the network parameters by maximizing the data likelihood formulated in terms of the linear combinations of the integral networks. Finally, we reassemble the parameters of the integral network to obtain the intensity. This approach leads to the exact intensity function and its antiderivative without restricting its parametric forms, thereby making the integration process "automatic".

Our approach bears a resemblance with the fully NN approach by Omi et al. [2019] for 1D temporal point processes. There, automatic integration can be easily implemented by imposing monotonicity constraints on the integral network [Lindell et al., 2021, Li et al., 2019]. However, due to triple integration in STPP, imposing monotonicity constraints significantly hurdles the expressivity of the integral network, leading to inaccurate intensity. As the experiments show, extending Fully NN to 3D cannot learn complex spatiotemporal intensity functions. Instead, we propose a decomposable parametrization for the integral network that bypasses this restriction.

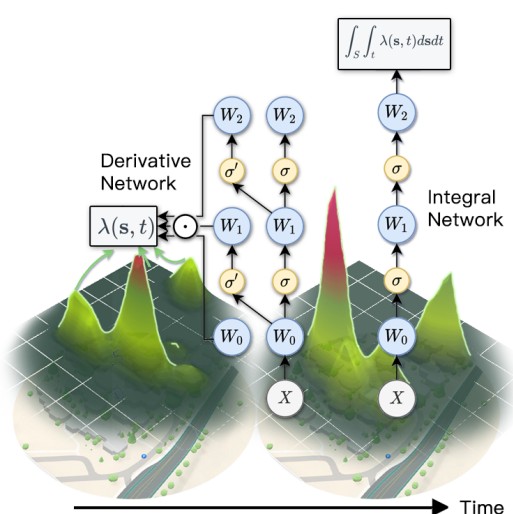

Figure 1: Illustration of `AutoSTPP`. $W$ denotes the linear layer's weight. $\sigma$ is the nonlinear activation function. Left shows the intensity network approximating $\lambda(\mathbf{s}, t)$, and right is the integral network that computes $\int_{\mathbf{s}} \int_t \lambda$. The two networks share the same parameters.

Our approach can efficiently compute the exact likelihood of *any* continuous influence function. We validate our approach using synthetic spatiotemporal point processes with complex intensity functions. We also demonstrate the superior performance on several real-world discrete event datasets. Compared to FullyNN by Omi et al. [2019], our approach presents a more general higher-order automatic integration scheme and is more effective in learning complex intensity functions. Also, the probability density of Omi et al. [2019] is ill-defined as its intensity integral does not diverge to infinity [Shchur et al., 2019]. We fix the issue by adding a constant background intensity $\mu$.

To summarize, our contributions include:

- We propose the first deep learning framework `AutoSTPP` to speed up spatiotemporal point process learning with automatic integration. We use dual networks and enforce the non-negativity of the intensity via a monotone integral network.
- We show that our automatic integration scheme empirically learns intensity functions more accurately than other integration approaches.
- We prove that the derivative network of `AutoSTPP` is a universal approximator of continuous functions and, therefore, is a consistent estimator under mild assumptions.

- We demonstrate that `AutoSTPP` can recover complex influence functions from synthetic data, enjoys high training efficiency and model interpretability, and outperforms the state-of-the-art methods on benchmark real-world data.

## 2 Related Work

**Parametrizing Point Process.** Fitting traditional STPP, such as the spatiotemporal Hawkes process, to data points with parametric models can perform poorly if the model is misspecified. To address this issue, statisticians have extensively studied semi- and non-parametric inference for STPP. Early works like Brix and Moller [2001], Brix and Diggle [2001] usually rely on Log-Gaussian Cox processes as a backbone and Epanechnikov kernels as estimators of the pair correlation function. Adams et al. [2009] propose a non-parametric approach that allows the generation of exact Poisson data from a random intensity drawn from a Gaussian Process, thus avoiding finite-dimensional approximation. These Cox process models are scalable but assume a continuous intensity change over time.

Recently, neural point processes (NPPs) that combine point processes with neural networks have received considerable attention [Yan et al., 2018, Upadhyay et al., 2018, Huang et al., 2019, Shang and Sun, 2019, Zhang et al., 2020]. Under this framework, models focus more on approximating a discrete set of intensities before and after each event. The continuous intensity comes from interpolating the intensities between discrete events. For example, [Du et al., 2016] uses an RNN to generate intensities after each event. [Mei and Eisner, 2016] proposes a novel RNN architecture that generates intensities at both ends of each inter-event interval. Other works consider alternative training schema: Xiao et al. [2017] used Wasserstein distance, Guo et al. [2018] introduced noise-contrastive estimation, and Li et al. [2018] leveraged reinforcement learning. While these NPP models are more expressive than the traditional point process models, they still assume simple (continuous, usually monotonous) inter-event intensity changes and only focus on temporal point processes.

Neural STPP [Chen et al., 2020, Zhou et al., 2022] further generalizes NPP to spatiotemporal data. They use a non-negative activation function to map the hidden states to a scalar, i.e., the temporal intensity immediately after an event, and a conditional spatial distribution. The change of intensity between events is represented by a decay function or a Neural ODE. The conditional spatial distribution is represented by a kernel mixture or a normalizing flow. Nevertheless, all models assume a continuous transformation of the intensity function and have limited expressivity.

In the context of temporal point processes (TPP), closely related approaches are Omi et al. [2019] and Zhou and Yu [2023]. Both propose using a Deep Neural Network (DNN) to parameterize the integral of an intensity function. The work by Omi et al. [2019] offers a more flexible formulation yet does not incorporate any specific prior assumptions about the form of the intensity changes. On the other hand, the approach by Zhou and Yu [2023] is capable of capturing more sophisticated influence functions, and it is this work that our research builds upon. However, both of these studies focus on the easier problem of learning the derivative with respect to time alone, thereby neglecting the rich amount of other features that may associated with the timestamps.

**Integration Methods.** Integration methods are largely ignored in NPP literature but are central to a model's ability to capture the complex dynamics of a system. Existing works either use an intensity function with an elementary integral [Du et al., 2016] or Monte Carlo integration [Mei and Eisner, 2016]. However, we will see in the experiment section that the choice of integration method has a non-trivial effect on the model performance.

Integration is generally more complicated than differentiation, which can be mechanically solved using the chain rule. Most integration rules, e.g., integration by parts and change of variables, transform an antiderivative to another that is not necessarily easier. Elementary antiderivative only exists for a small set of functions, but not for simple composite functions such as $\exp(x^2)$ [Dunham, 2018]. The Risch algorithm can determine such elementary antiderivative [Risch, 1969, 1970] but has never been fully implemented due to its complexity. The most commonly used integration methods are still numerical: Newton-Cotes Methods, Romberg Integration, Quadrature, and Monte Carlo integration [Davis and Rabinowitz, 2007].

Several recent works leverage automatic differentiation to speedup integration, a paradigm known as Automatic Integration (AutoInt). Liu [2020] proposes integrating the Taylor polynomial using the derivatives from Automatic Differentiation (AutoDiff). It requires partitioning of the integral limits

and choosing the order of Taylor approximation. Though it makes use of AutoDiff, the integration procedure involves a trade-off between runtime and accuracy and is numerical in nature. Li et al. [2019] and Lindell et al. [2021] proposed a dual network approach, which we will discuss in detail in Section 3. This approach guarantees a closed-form integral and is efficient.

## 3 Methodology

We first review the background of Spatiotemporal Point Processes. Then, we introduce the AutoInt technique, which is interpretable and flexible. Lastly, we consider applying the AutoInt technique to the 3D integration in the spatiotemporal point process.

### 3.1 Spatiotemporal Point Process

A spatiotemporal point process (STPP) generalizes TPP to model the number of events $N(\mathcal{S} \times (a, b))$ that occurred in the Cartesian product of the spatial domain $\mathcal{S} \subseteq \mathbb{R}^d$ ($d$ is the spatial dimensionality) and the time interval $(a, b]$. It is characterized by a non-negative *space-time intensity function* given the event history $\mathcal{H}_t := \{(\mathbf{s}_1, t_1), \ldots, (\mathbf{s}_n, t_n)\}_{t_n \leq t}$,

$$\lambda^*(\mathbf{s}, t) := \lim_{\Delta \mathbf{s} \to 0, \Delta t \to 0} \frac{\mathbb{E}[N(B(\mathbf{s}, \Delta \mathbf{s}) \times (t, t + \Delta t))|\mathcal{H}_t]}{B(\mathbf{s}, \Delta \mathbf{s})\Delta t} \tag{2}$$

which is the probability of finding an event in an infinitesimal time interval $(t, t + \Delta t]$ and an infinitesimal spatial ball $\mathcal{S} = B(\mathbf{s}, \Delta \mathbf{s})$ centered at location $\mathbf{s}$. Alternatively, an STPP can be seen as a TPP with a conditional spatial distribution $f^*(\mathbf{s}|t)$, such that $\lambda^*(\mathbf{s}, t) = \lambda^*(t)f^*(\mathbf{s}|t)$.

### 3.2 AutoInt Point Process

Consider the following NPP model that generalizes the spatiotemporal Hawkes process:

$$\lambda^*(\mathbf{s}, t) = \mu + \sum_{t_i < t} f_\theta^+(\mathbf{s} - \mathbf{s}_i, t - t_i, \mathcal{H}(t_i)). \tag{3}$$

Here $\mu$ is the base intensity. $f_\theta^+$ is a positive scalar function that takes space, time, and representations of event history $\mathcal{H}(\mathbf{s}_i, t_i)$ as inputs. Each $f_\theta^+$ is approximated by a DNN.

The two main benefits of such design are flexibility and interpretability. $f_\theta$ is a neural network that can model complex inter-event change. The additive form allows the decomposition of the intensity function for event influence analysis. We extend this model into the spatiotemporal domain,

### 3.3 Automatic Integration (AutoInt)

One advantage of the neural STPP model in Equation 3 is that we can instantiate automatic integration (AutoInt) and calculate the volume integral $\int_{t=a}^{b} f_\theta(\mathbf{s}, t, \mathbf{h}) := F_\theta(b, \mathbf{h}) - F_\theta(a, \mathbf{h})$, where $\mathbf{h}$ is the latent representation of the event history until $t$ generated by a deep sequence model. AutoInt first constructs the integral network $F_\theta$ and then reorganizes the computational graph of $F_\theta$ to form the integrant, the derivative network $f_\theta$. The two networks thus share the same set of parameters $\theta$.

Specifically, let $\mathbf{x} := \mathbf{s} \oplus t \oplus \mathbf{h}$, we approximate the integral of the intensity function with a DNN of the following form:
$$F_\theta(\mathbf{x}) = \mathbf{W}_n \cdots (\mathbf{W}_3 \sigma(\mathbf{W}_2 \sigma(\mathbf{W}_1 \mathbf{x}))),$$
where $n$ is the number of layers, $\mathbf{W}_k : \mathbb{R}^{M_k} \mapsto \mathbb{R}^{N_k}$ denotes the weight of the $k$-th linear layer of the neural network and $\sigma$ denotes the elementwise nonlinearity. $M_k$ and $N_k$ are the input and output dimension for the $k$-th layer. Hence, the set of parameters in this neural network is $\theta = \{\mathbf{W}_k \in \mathbb{R}^{M_k \times N_k}, \forall k\}$.

The derivative network $f_\theta$ is the partial derivative of the integral network $F_\theta$. As long as the activation function is differentiable everywhere, one can compute the intensity recursively,

$$f_\theta(\mathbf{x}) := \frac{\partial F_\theta}{\partial t}(\mathbf{x}) = \mathbf{W}_n \sigma'(\mathbf{W}_{n-1} \sigma(\mathbf{W}_{n-2} \ldots (\mathbf{W}_1 \mathbf{x}))) \cdots \circ \mathbf{W}_2 \sigma'(\mathbf{W}_1 \mathbf{x}) \circ \mathbf{W}_{11},$$

where $\circ$ indicates the Hadamard product, and $\mathbf{W}_{11}$ is the first column of $\mathbf{W}_1$, i.e.,

$$\mathbf{W}_1 := [\mathbf{W}_{11} \quad \mathbf{W}_{12} \quad \ldots \quad \mathbf{W}_{1,M_1}]$$

Computing $f_\theta(\mathbf{x})$ involves many repeated operations. For example, the result of $\mathbf{W}_1\mathbf{x}$ is used for compute both $\sigma(\mathbf{W}_1\mathbf{x})$ and $\sigma'(\mathbf{W}_1\mathbf{x})$, see Figure 1. We have developed a program that harnesses the power of dynamical programming to compute derivatives efficiently with AutoDiff. See Appendix D for the detailed algorithm.

## 3.4 AutoInt Point Processes as Consistent Estimators

We show that the universal approximation theorem (UAT) holds for derivative networks. This theorem signifies that, given a sufficient number of hidden units, derivative networks can theoretically approximate any continuous functions, no matter how complex. Therefore, using derivative networks does not limit the range of influence functions that can be approximated.

**Proposition 3.1** (Universal Approximation Theorem for Derivative Networks). *The set of derivative networks corresponding to two-layer feedforward integral networks is dense in $C(\mathbb{R})$ with respect to the uniform norm.*

For a detailed proof, see Appendix E. With UAT, it is clear that under some mild assumptions, AutoInt Point Processes are consistent estimators of point processes that take the form of Equation 5.

**Proposition 3.2** (Consistency of AutoInt Point Process). *Under the assumption that the ground truth point process is stationary, ergodic, absolutely continuous and predictable, if the ground truth influence function is truncated (i.e., $\exists C, f(t) = 0 \ \forall t > c$), the maximum likelihood estimator $f_\theta$ converges to the true influence function $f$ in probability as $T \to \infty$.*

Our model belongs to the class of linear self-excitation processes, whose maximum likelihood estimator properties were analyzed by Ogata et al. [1978]. Under the assumptions above, two conditions are needed for the proof of consistency:

**Assumption 3.3.** (Consistency Conditions) For any $\theta \in \Theta$ there is a neighbourhood $U$ of $\theta$ such that

1. $\sup_{\theta' \in U} |\lambda_{\theta'}(t, \omega) - \lambda^*_{\theta'}(t, \omega)| \to 0$ in probability as $t \to \infty$,

2. $\sup_{\theta' \in U} |\log \lambda^*_{\theta*}(t, \omega)|$ has, for some $\alpha > 0$, finite $(2 + \alpha)$ th moment uniform bounded with respect to $t$.

The first condition is satisfied by UAT. The second condition depends on the rate of decrease of the influence tail and is satisfied by truncation. In our experiments, we truncated the history by only including the influences of the previous 20 events. If the ground truth influence function decays over the entire time domain, our estimator may exhibit negligible bias.

## 3.5 3D Automatic Integration

AutoInt gives us an efficient way to calculate a line integral over one axis. However, for spatiotemporal point process models, we need to calculate the triple integral of the intensity function over space and time. Since we cannot evaluate the integral network with an input of infinity, we assume the spatial domain to be a rectangle, such that the triple integral is over a cuboid. We then convert the triple integral to line integrals using Divergence and Green's theorem [Marsden and Tromba, 2003].

Define $\mathbf{s} := (x, y)$ and $t := z$, we model the spatiotemporal influence $f^*_\theta(x, y, z)$ as

$$f^*_\theta(\mathbf{s}, t) = \frac{\partial P}{\partial x} + \frac{\partial Q}{\partial y} + \frac{\partial R}{\partial z} \tag{4}$$

The Divergence theorem relates volume and surface integrals. It states that for any $P(x, y, z), Q(x, y, z), R(x, y, z)$ that is differentiable over the cuboid $\Omega$, we have

$$\iiint_\Omega \left( \frac{\partial P}{\partial x} + \frac{\partial Q}{\partial y} + \frac{\partial R}{\partial z} \right) dv = \oiint_\Sigma P dy dz + Q dz dx + R dx dy,$$

where $\Sigma$ are the six rectangles that enclose $\Omega$.

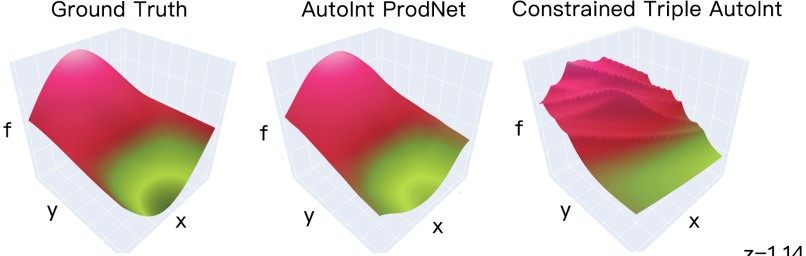

Figure 2: Comparison of fitting a nonnegative function $f(x, y, z) = \sin(x)\cos(y)\sin(z) + 1$. Our proposed AutoInt ProdNet approach can well approximate the ground truth function. In contrast, imposing the constraint through activation function with a nonnegative triple derivative "Constrained Triple AutoInt" fails due to overly stringent constraint.

Using $R$ as an example, we begin with two neural network approximators, $L_R$ and $M_R$. We initialize the two approximators as first-order derivative networks, whose corresponding integral networks are $\int L_R dx$ and $\int M_R dy$. We use the $xy$ partial derivatives of the two networks to model $R$, such that $R := \dfrac{\partial M_R}{\partial x} - \dfrac{\partial L_R}{\partial y}$. Then the $z$ partial derivative $\dfrac{\partial R}{\partial z}$ is $\dfrac{\partial^2 M_R}{\partial x \partial z} - \dfrac{\partial^2 L_R}{\partial y \partial z} = \dfrac{\partial^3 \int M_R dy}{\partial x \partial y \partial z} - \dfrac{\partial^3 \int L_R dx}{\partial x \partial y \partial z}$. We can see that $\dfrac{\partial R}{\partial z}$ can be exactly evaluated as the third derivatives of two integral networks. In the same manner, we can model $P$ and $Q$ such that the intensity is parametrized by six networks: $L_R, L_Q, L_P, M_R, M_Q$, and $M_P$.

We use neural network pairs $\{L, M\}$ to parametrize each intensity term in Equation 4 according to Green's theorem. It states that for any $L$ and $M$ differentiable over the rectangle $D$,

$$\iint_D \left( \frac{\partial L}{\partial x} - \frac{\partial M}{\partial y} \right) dx dy = \oint_{L^+} (L dx + M dy),$$

where $L^+$ is the counterclockwise path that encloses $D$. $\int R dx dy$ is then $\oint L_R dx + M_R dy$, and can be exactly evaluated using the integral networks.

In practice, we observe that the six networks' parametrization of intensity can be simplified. $L_R, L_Q, L_P$ can share the same set of weights, and similarly for $M_R, M_Q, M_P$. The influence is essentially parametrized by two different neural networks, $L$ and $M$. That is,

$$f_\theta(t, \mathbf{h}) := \left( \frac{\delta^3 \int M dy}{\delta x \delta y \delta z} - \frac{\delta^3 \int L dx}{\delta x \delta y \delta z} \right) + \left( \frac{\delta^3 \int M dz}{\delta x \delta y \delta z} - \frac{\delta^3 \int L dy}{\delta x \delta y \delta z} \right) + \left( \frac{\delta^3 \int M dx}{\delta x \delta y \delta z} - \frac{\delta^3 \int L dz}{\delta x \delta y \delta z} \right)$$

### 3.6 Imposing the 3D Non-negativity Constraint

For the NPP model in Equation 3, the derivative network $f_\theta$ needs to be non-negative. Imposing the non-negativity constraint for 3D AutoInt is a challenging task. It implies that the integral network $F_\theta$ always has a nonnegative triple derivative.

A simple approach is to apply an activation function with a nonnegative triple derivative. An integral network that uses this activation and has nonnegative linear layer weights satisfies the condition. We call this approach "Constrained Triple AutoInt". However, the output of an integral network can grow very quickly with large input, and the gradients are likely to explode during training. Moreover, the non-negative constraint on the influence function only requires $\dfrac{\partial F_\theta}{\partial \mathbf{s} \partial t}$ to be positive. But an activation function with a nonnegative triple derivative would also enforce other partial derivatives to be positive. Such constraints are overly restrictive for STPPs, whose partial derivatives $\dfrac{\partial F_\theta}{\partial \mathbf{s} \partial \mathbf{s}}$ and $\dfrac{\partial F_\theta}{\partial t \partial t}$ can both be negative when the intensity is well-defined.

**ProdNet.** We propose a different solution to enforce the 3D non-negative constraint called *ProdNet*. Specifically, we decompose the influence function $f_\theta(s_1, s_2, t) : \mathbb{R}^3 \to \mathbb{R}$ as $f_\theta^1(s_1) f_\theta^2(s_2) f_\theta^3(t)$, the product of three $\mathbb{R} \to \mathbb{R}$ AutoInt derivative networks. The triple antiderivative of the influence

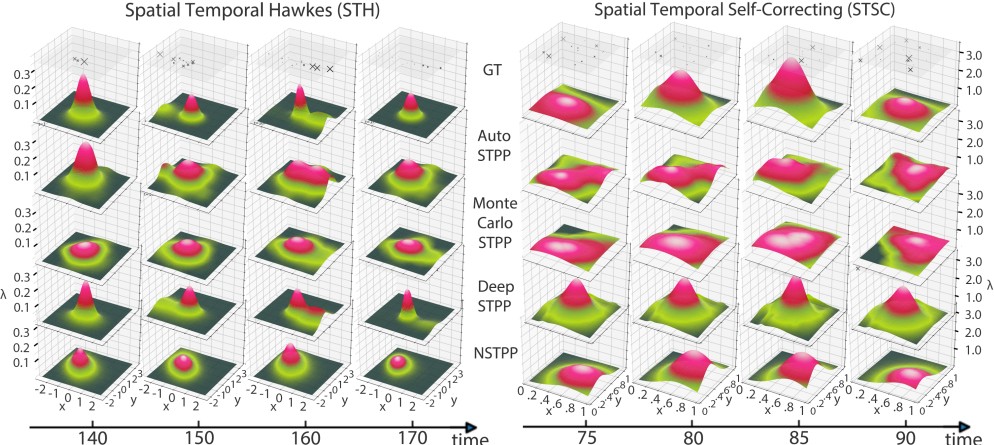

Figure 3: Comparing the ground truth conditional intensity $\lambda^*(\mathbf{s}, t)$ with the learned intensity on the *ST Hawkes* Dataset 1 and *ST Self-Correcting* Dataset 3. **Top row:** Ground truth. **Second row:** Our `AutoSTPP`. **Rest of the rows:** Baselines. The crosses on top represent past events. Larger crosses indicate more recent events.

function is then $F_\theta^1(s_1)F_\theta^2(s_2)F_\theta^3(t)$, the product of their respective integral networks. Then we can apply 1D non-negative constraint to each of the derivative networks. The other partial derivatives are not constrained because $F_\theta^1, F_\theta^2, F_\theta^3$ can be negative.

One limitation of such decomposition is that it can only learn the joint density of marginally independent distribution. We circumvent this issue by parameterizing the influence function as the sum of $N$ ProdNet, $\sum_{i=1}^N f_{\theta,i}^1(s_1)f_{\theta,i}^2(s_2)f_{\theta,i}^3(t)$. The formulation is no longer marginally independent since it is not multiplicative decomposable. Figure 2 shows that the sum of two ProdNets is already sufficient to approximate a function that cannot be written as the sum of products of positive functions. In contrast, the constrained triple AutoInt's intensity is convex everywhere and fails to fit $f$.

Increasing the number of ProdNet improves AutoInt's flexibility at the cost of time and memory. We perform an ablation study of the number of ProdNet in Appendix F.

### 3.7 Model Training

Given the integral network $F_\theta(t, \mathbf{h}) := \int_{\mathcal{S}} F_\theta(\mathbf{s}, t, \mathbf{h})$ and the derivative network approximating the influence function $f_\theta = \frac{\partial F_\theta}{\partial t \partial \mathbf{s}}$, the log-likelihood of an event sequence $\mathcal{H}_n = \{(\mathbf{s}_1, t_1), \cdots, (\mathbf{s}_n, t_n)\}$ observed in time interval $[0, T]$ with respect to the model is

$$\mathcal{L}(\mathcal{H}_n) = \sum_{i=1}^n \log \left( \sum_{j=1}^{i-1} f_\theta(\mathbf{s}_i - \mathbf{s}_j, t_i - t_j, \mathbf{h}_i) \right) - \sum_{i=1}^n \left( F_\theta(T - t_i, \mathbf{h}_n) - F_\theta(0, \mathbf{h}_i) \right)$$

Obtaining the above from Equation 1 is straightforward by the Fundamental Theorem of Calculus. $F_\theta$ is evaluated using the Divergence theorem. We can learn the parameters $\theta$ in both networks by maximizing the log-likelihood function. In experiments, we parametrize $f_\theta$ with two AutoInt networks $L$ and $M$. Each network is a sum of $N$ ProdNets. That is,

$$f_\theta(\mathbf{s}, t) := \sum_{i=1}^N f_{\theta_i}(\mathbf{s}, t) = 3 \sum_{i=1}^N \prod_{x \in \{s_1, s_2, t\}} \left[ \frac{\delta^3 \int M_i dx}{\delta x^3} - \frac{\delta^3 \int L_i dx}{\delta x^3} \right]$$

We name our method Automatic Spatiotemporal Point Process (`AutoSTPP`).

## 4 Experiments

We compare the performances of different neural STPPs using synthetic and real-world benchmark data. For synthetic data, our goal is to validate our `AutoSTPP` can accurately recover complex

Table 1: Test log likelihood (LL) and Hellinger distance of distribution (HD) on synthetic data (LL higher is better, HD lower is better). Comparison between AutoSTPP, NSTPP, Monte Carlo STPP, on synthetic datasets from two types of spatiotemporal point processes.

| | Spatiotemporal Hawkes process | | | | | | Spatiotemporal Self Correcting process | | | | | |
| | DS1 | | DS2 | | DS3 | | DS1 | | DS2 | | DS3 | |
| | LL | HD | LL | HD | LL | HD | LL | HD | LL | HD | LL | HD |
| NSTPP | -5.3110 | 0.5341 | -4.8564 | 0.5849 | -3.7366 | 0.1498 | -2.0759 | 0.5426 | -2.3612 | 0.3933 | -3.0599 | 0.3097 |
| DSTPP | **-3.8240** | **0.0033** | -3.1142 | 0.4920 | **-3.6327** | **0.0908** | -1.2248 | 0.2348 | -1.4915 | 0.1813 | **-1.3927** | **0.2075** |
| Monte Carlo STPP | -4.0066 | 0.3198 | -3.2778 | 0.3780 | -3.7704 | 0.2587 | -1.0317 | 0.1224 | -1.3681 | 0.1163 | -1.4439 | 0.2334 |
| AutoSTPP | -3.9548 | 0.3018 | **-2.5304** | **0.1891** | -3.7700 | 0.1495 | **-1.0269** | **0.1216** | **-1.3657** | **0.1119** | -1.3979 | 0.2181 |

intensity functions. Additionally, we show that the errors resulting from numerical integration lead to a higher variance in the learned intensity than closed-form integration. We show that our model performs better or on par with the state-of-the-art methods for real-world data.

## 4.1 Experimental Setup

**Synthetic Datasets.** We follow the experiment design of Zhou et al. [2022] to validate that our method can accurately recover the true intensity functions of complex STPPs. We use six synthetic point process datasets simulated using Ogata's thinning algorithm [Chen, 2016], see Appendix B for details. The first three datasets were based on spatiotemporal Hawkes processes, while the remaining three were based on spatiotemporal self-correcting processes. Each dataset spans a time range of $[0, 10000)$ and is generated using a fixed set of parameters. Each dataset was divided into a training, validation, and testing set in an $8 : 1 : 1$ ratio based on the time range.

**Spatiotemporal Hawkes process (STH).** A spatiotemporal Hawkes process, also known as a self-exciting process, posits that every past event exerts an additive, positive, and spatially local influence on future events. This pattern is commonly observed in social media and earthquakes. The process is characterized by the intensity function [Reinhart, 2018]:

$$\lambda^*(\mathbf{s}, t) := \mu g_0(\mathbf{s}) + \sum_{i:t_i<t} g_1(t, t_i) g_2(\mathbf{s}, \mathbf{s}_i) : \mu > 0. \tag{5}$$

$g_0$ represents the density of the background event distribution over $\mathcal{S}$. $g_2$ represents the density of the event influence distribution centered at $\mathbf{s}_i$ and over $\mathcal{S}$. $g_1$ describes each event $t_i$'s influence decay over time. We implement $g_0$ and $g_2$ as Gaussian densities and $g_2$ as exponential decay functions.

**Spatiotemporal Self-Correcting process (STSC).** A spatiotemporal Self-Correcting process assumes that the background intensity always increases between the event arrivals. Each event discretely reduces the intensity in the vicinity. The STSC is often used for modeling events with regular intervals, such as animal feeding times. It is characterized by:

$$\lambda^*(\mathbf{s}, t) = \mu \exp\left(g_0(\mathbf{s})\beta t - \sum_{i:t_i<t} \alpha g_2(\mathbf{s}, \mathbf{s}_i)\right) : \alpha, \beta, \mu > 0 \tag{6}$$

$g_0(\mathbf{s})$ again represents the density of the background event distribution. $g_2(\mathbf{s}, \mathbf{s}_i)$ represents the density of the negative event influence centered at $\mathbf{s}_i$.

See the Appendix B for the simulation parameters of the six synthetic datasets.

**Real-world Datasets.** We follow the experiment design of Chen et al. [2020] and use two of the real-world datasets, *Earthquake Japan* and *COVID New Jersey*. The first dataset includes information on the times and locations of all earthquakes in Japan between 1990 and 2020, with magnitudes of at least 2.5. This dataset includes 1050 sequences over a $[0, 30)$ time range and is split with a ratio of $950 : 50 : 50$. The second dataset is published by The New York Times and describes COVID-19 cases in New Jersey at the county level. This dataset includes 1650 sequences over a $[0, 7)$ time range and is split with a ratio of $1450 : 100 : 100$.

**Evaluation Metrics.** For real-world datasets, we report the average test log-likelihood (LL) of events. For synthetic datasets, the ground truth intensities are available, so we report the test log-likelihood

(LL) and the time-average Hellinger distance between the learned conditional spatial distribution $f^*(\mathbf{s}|t)$ and the ground truth distribution. The distributions are estimated as multinomial distributions $P = \{p_i, ..., p_k\}$ and $Q = \{q_i, ..., q_k\}$ at $k$ discretized grid points. The Hellinger distance is then calculated as $H(P, Q) = \frac{1}{\sqrt{2}} \sqrt{\sum_{i=1}^{k} \left( \sqrt{p_i} - \sqrt{q_i} \right)^2}$

**Baselines.** We compare with two state-of-the-art neural STPP models, NSTPP [Chen et al., 2020] and Deep-STPP [Zhou et al., 2022]. We also design another baseline, Monte Carlo STPP, which uses the same underlying model as `AutoSTPP` but applies numerical integration instead of AutoInt to calculate the loss. For a fair comparison, Monte Carlo STPP models the influence $f_\theta^*(s, t)$ as a multi-layer perceptron instead of a derivative network. This numerical baseline aims to demonstrate the benefit of automatic integration.

## 4.2 Results and Discussion

**Exact Likelihood.** Not all baseline methods have exact likelihood available. NSTPP [Chen et al., 2020] uses a numerical Neural-ODE solver. DeepSTPP [Zhou et al., 2022] introduces a VAE that optimizes a lower bound of likelihood. Monte Carlo STPP estimates the triple integral of intensity by Monte Carlo integration. As such, the results presented in Table 1 are estimated for these baseline models, whereas for our model, the results reflect the true likelihood.

**Advantage of AutoInt.** We visualize and compare two sample sets of intensities from STH Dataset 1 and STSC Dataset 3 in Figure 3. While the influence function approximator in Monte Carlo STPP can theoretically approximate more functions than Auto-STPP, the intensity it learns is "flatter" than the intensity learned by Auto-STPP. This flatness indicates a lack of information regarding future event locations.

Figure 4: Forward mixed $(d/dx_1 dx_2 \cdots dx_k)$ partial derivative computation average speed comparison between our efficient implementation and PyTorch naive AutoGrad, for a two-layer MLP.

The primary reason behind this flatness can be traced back to numerical errors inherent in the Monte Carlo method. The Monte Carlo integration performs poorly for sharply localized integrants because its samples are homogeneous over the phase space. The intensity of ST-Hawkes is a localized function; it is close to zero in most of the places but high near some event locations. As a result, Monte Carlo STPP can hardly recover the sharpness of the ground truth intensity. NSTPP also uses Monte Carlo integration and suffers the same drawback on STH Dataset 1. In contrast, `AutoSTPP` evaluates the integral with closed-form integration and alleviates this issue.

**Synthetic Datasets Results.** Table 1 compares the test LL and the Hellinger distance between `AutoSTPP` and the baseline models on the six synthetic datasets. For the STSC datasets, we can see that `AutoSTPP` accurately recovers the intensity functions compared to other models. In Figure 3, `AutoSTPP` is the only model whose peak intensity location is always the same as the ground truth. DeepSTPP does not perform well in learning the dynamics of the STSC dataset; it struggles to align the peak and tends to learn the flat intensity as overly sharp. The peak intensity location of NSTPP is also biased.

Table 2: Test log likelihood (LL) comparison for space and time on real-world benchmark data, mean and standard deviation over three runs.

| LL | COVID-19 NY | Earthquake JP |
|---|---|---|
| NSTPP | $2.5566_{\pm 0.0447}$ | $-4.4949_{\pm 0.1172}$ |
| DSTPP | $2.3433_{\pm 0.0109}$ | $-3.9852_{\pm 0.0129}$ |
| MonteSTPP | $2.1070_{\pm 0.0342}$ | $-3.6085_{\pm 0.0436}$ |
| AutoSTPP | $\mathbf{2.6243}_{\pm 0.5905}$ | $\mathbf{-3.5948}_{\pm 0.0025}$ |

For the STHP datasets, DeepSTPP has a clear advantage because it uses Gaussian kernels to approximate the Gaussian ground truth. In Table 1, `AutoSTPP` outperforms all other models except DeepSTPP, and its performance is comparable. Figure 3 shows that Monte Carlo STPP and NSTPP

can only learn an unimodal function, whereas `AutoSTPP` can capture multi-modal behavior in the ground truth, especially the small bumps near the mode.

**Real-world Datasets Results.** Table 2 compares the test LL of `AutoSTPP` against the baseline models on the earthquakes and COVID datasets. Our model demonstrates superior performance, outperforming all the state-of-the-art methods. One should note that while Monte Carlo STPP shows performance comparable to ours on the Earthquake JP dataset, it falls short when applied to the COVID-19 NY dataset. We attribute this discrepancy to the large low-population-density areas in the COVID-19 NY data, which causes higher numerical error in integration.

### 4.3 Computational Efficiency

Figure 4 visualizes the benefit of using our implementation of AutoInt instead of the PyTorch naive implementation. More visualizations of the forward and backward computation times can be found in Appendix A. We can see that our implementation can be extended to compute any order of partial derivative. It is significantly faster than the naive autograd. In our AutoSTPP, we calculate the intensity using the product of three first-order derivatives. Our implementation would lead to a speedup of up to 68% for computing each first-order derivative.

## 5 Conclusion

We propose Automatic Integration for neural spatiotemporal point process models (`AutoSTPP`) using a dual network approach. `AutoSTPP` can efficiently compute the exact likelihood of *any* sophisticated intensity.

We validate the effectiveness of our method using synthetic data and real-world datasets and demonstrate that it significantly outperforms other point process models with numerical integration when the ground truth intensity function is localized.

However, like any approach, ours is not without its limitations. While `AutoSTPP` excels in computing the likelihood, sampling from `AutoSTPP` is computationally expensive as we only have the expression of the probability density. Closed-form computation of expectations is also not possible; Knowing the form of $\int \lambda(t)$, calculating $\int t\lambda(t)$ is still intractable.

Our work presents a new paradigm for learning continuous-time dynamics. Currently, our neural process model takes the form of Hawkes processes (self-exciting) but cannot handle the discrete decreases of intensity after events due to the difficulty of integration. Future work includes relaxing the form of the intensity network with advanced integration techniques. Another interesting direction is to increase the approximation ability of the product network.

## Acknowledgement

This work was supported in part by U. S. Army Research Office under Army-ECASE award W911NF-07-R-0003-03, the U.S. Department Of Energy, Office of Science, IARPA HAYSTAC Program, NSF Grants #2205093, #2146343, and #2134274.

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

# A   More Implementation Benchmarks

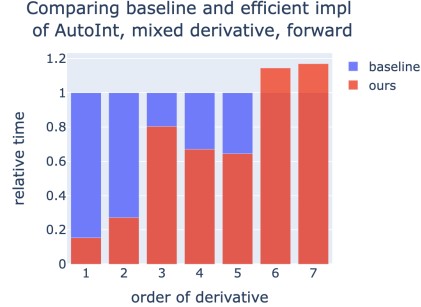

Figure 5: Forward mixed ($d/dx_1 dx_2...dx_k$) partial derivative computation speed, 3 layers MLP

Figure 6: Forward mixed ($d/dx_1 dx_2...dx_k$) partial derivative computation speed, 4 layers MLP

As the number of MLP layers increases, the lower-order derivative computation becomes faster (relative to PyTorch naive implementation), whereas the higher-order derivative computation becomes slower. This performance pattern is because our implementation uses Python for loop. Our approach is faster than the baseline in the majority of cases.

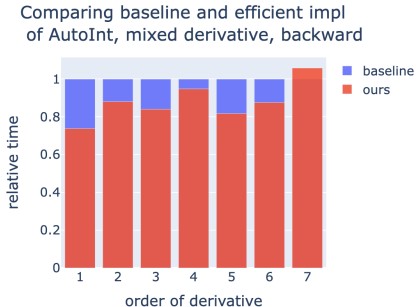
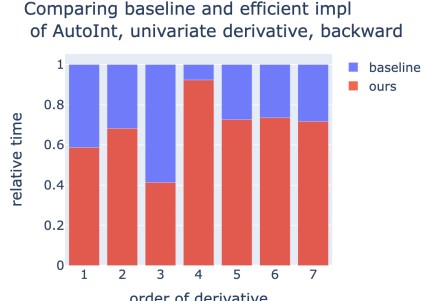

Figure 7: Forward + Backward mixed ($d/dx_1 dx_2 \cdots dx_k$) partial derivative computation speed, 3 layers MLP

Figure 8: Forward + Backward univariate ($d/dx_1 dx_1 \cdots dx_1$) partial derivative computation speed, 3 layers MLP

As for both forward and backward computation times, our implementation still consistently surpasses the baseline in terms of speed. Notably, our method is even more efficient when calculating univariate partial derivatives than mixed partial derivatives. This advantage is primarily due to the reduced number of iterations required by the Python for-loop in the case of univariate derivatives.

# B   STSC and ST-Hawkes Introduction and Simulation Parameters

We use the same parameters as Zhou et al. [2022].

The STSCP's and the STHP's kernels $g_0(\mathbf{s})$ and $g_2(\mathbf{s}, \mathbf{s}_j)$ are prespecified to be Gaussian:

$$g_0(\mathbf{s}) := \frac{1}{2\pi} |\Sigma_{g0}|^{-\frac{1}{2}} \exp\left(-\frac{1}{2}(\mathbf{s} - [0,0])\Sigma_{g0}^{-1}(\mathbf{s} - [0,0])^T\right)$$

$$g_2(\mathbf{s}, \mathbf{s}_j) := \frac{1}{2\pi} |\Sigma_{g2}|^{-\frac{1}{2}} \exp\left(-\frac{1}{2}(\mathbf{s} - \mathbf{s}_j)\Sigma_{g2}^{-1}(\mathbf{s} - \mathbf{s}_j)^T\right)$$

The STSCP is defined on $\mathcal{S} = [0,1] \times [0,1]$, while the STHP is defined on $\mathcal{S} = \mathbb{R}^2$. The STSCP's kernel functions are normalized according to their cumulative probability on $\mathcal{S}$. Table 4 shows the

simulation parameters. We discretized the STSCP's spatial domain as a $101 \times 101$ grid during the simulation.

Table 3: Parameter settings for the synthetic dataset

|  |  | $\alpha$ | $\beta$ | $\mu$ | $\Sigma_{g0}$ | $\Sigma_{g2}$ |
|---|---|---|---|---|---|---|
| ST-Hawkes | DS1 | .5 | 1 | .2 | [.2 0; 0 .2] | [0.5 0; 0 0.5] |
|  | DS2 | .5 | .6 | .15 | [5 0; 0 5] | [.1 0; 0 .1] |
|  | DS3 | .3 | 2 | 1 | [1 0; 0 1] | [.1 0; 0 .1] |
| ST-Self Correcting | DS1 | .2 | .2 | 1 | [1 0; 0 1] | [0.85 0; 0 0.85] |
|  | DS2 | .3 | .2 | 1 | [.4 0; 0 .4] | [.3 0; 0 .3] |
|  | DS3 | .4 | .2 | 1 | [.25 0; 0 .25] | [.2 0; 0 .2] |

Each dataset is a single, long sequence that spans over 10,000 time units. We divide each dataset into 50 sequences, each spanning 200 time units. We use 40 sequences for training, 5 for validation, and 5 for testing. Here's a summary of the total number of events found in each dataset:

Table 4: Number of events in each synthetic dataset

|  |  | number of events |
|---|---|---|
| ST-Hawkes | DS1 | 3983 |
|  | DS2 | 9017 |
|  | DS3 | 11693 |
| ST-Self Correcting | DS1 | 10002 |
|  | DS2 | 6668 |
|  | DS3 | 5004 |

The prediction task applies sliding windows to each of the datasets. We try to use historical events to predict the likelihood of the next event in the same sequence.

## C   Model Setup Details

We detail the specific hyperparameter settings in Table 5. Except for the learning rate, the same set of parameters was applied across all datasets. Despite varying datasets, This consistency in performance demonstrates our model's robustness to hyperparameters.

| Name | Value | Description |
|---|---|---|
| Optimizer | Adam | - |
| Learning rate | - | Depends on dataset, [0.0002, 0.004] |
| Momentum | 0.9 | Adam momentum |
| Epoch | 50 / 100 | 50 for synthetic dataset and 100 for real-world dataset |
| Batch size | 128 | - |
| Activation | tanh | Activation function in L and M (intensity parameter networks) |
| $N$ | 2 / 10 | Number of product nets to sum in L and M |
|  |  | 2 for synthetic dataset and 10 for real-world dataset |
| bias | true | L and M use bias in their linear layers |

Table 5: Hyperparameter settings for training `AutoSTPP` on all datasets.

## D   Forward-pass Algorithm for Automatic Integration

**Function:** `dnforward`$(f, n, x, \texttt{dims})$, `partition`$(n, k)$ finds all k-subset partitions of $n$

**Data:** $n$, dimension of $f(x)$, $x$, a tensor of shape (`batch`, `dim`),
`dims`, list of dimensions to derive, `layers`, composite functions in $f$
**Result:** $d^{\text{dims}} f / dx^{\text{dims}}$
Initialize dictionary `dnf`, mapping from `dims` to $d^{\text{dims}} f / dx^{\text{dims}}$, empty list `pd`
**if** $|dims| = 1$ **then**
    Precompute $f(x)$
    `pd` $\leftarrow \boldsymbol{\delta}_{i,\text{dim}}|_{i \in [1,n]}$
**else**
    **for** $subdims \in combination(dims, len(dims)\text{-}1)$ **do**
        Precompute `dnforward`$(f, n, x, \text{subdims})$
    **end**
    `pd` $\leftarrow \mathbf{0}$
**end**
**for** $layer \in layers$ **do**
    **if** $layer$ *is linear* **then**
        `pd` append last `pd` $\times W^T$, $W$ is the linear weight
    **else if** $layer$ *is activation* **then**
        **if** $|dims| = 1$ **then**
            `termsum` $\leftarrow$ last `pd` $\times$ `layer`$'(f)$
        **else**
            `termsum` $\leftarrow 0$
            **for** $order \in 0, \cdots, |dims|$ **do**
                **if** $order = 0$ **then**
                    `term` $\leftarrow$ last `pd`
                **else**
                    `term` $\leftarrow 0$
                    **for** $part \in partition(dims, order + 1)$ **do**
                        `temp` $\leftarrow 1$
                        **for** $subdims \in part$ **do**
                            `temp` $\leftarrow$ `temp` $\times$ `dnforward`$(f, n, x, \text{subdims})$ (precomputed)
                            `term` $\leftarrow$ `term` $+$ `temp`
                        **end**
                      `termsum` $\leftarrow$ `termsum` $+$ `term`
                    **end**
                **end**
                `termsum` $\leftarrow$ `termsum` $\times$ `layer`$^{(n)}(f)$
        **end**
    **end**
    `pd` append `termsum`
**end**
**return** last `pd`

## E  Universal Approximation Theorem for Derivative Network

Consider an AutoInt integral network with the form

$$g(x) = C \cdot (\sigma \circ (A \cdot x + b)), \qquad A \subseteq \mathbb{R}^{k \times n}, b \subseteq \mathbb{R}^k, C \subseteq \mathbb{R}^k,$$

where $\sigma$ denotes a $\mathbb{R} \to \mathbb{R}$ continuous non-polynomial function applied elementwise to each input dimension.

The derivative network thus takes the form

$$g'(x) = C \cdot (\sigma' \circ (A \cdot x + b) \circ A_{col}),$$

where $A_{col} \subseteq \mathbb{R}^k$ is a column of $A$ that corresponds to the deriving dimension.

Recall the universal approximation theorem [Daniels and Velikova, 2010], which says for every compact $K \subseteq \mathbb{R}^n$ and $f \in C(K, \mathbb{R}), \varepsilon > 0$, there exist $A, b, C$ such that

$$\sup_{x \in K} \| f(x) - g(x) \| < \varepsilon$$

**Proposition E.1.** *(Universal Approximation Theorem for Derivative Network) for every compact* $K \subseteq \mathbb{R}^n$ *and* $f \in C(K, \mathbb{R}), \varepsilon > 0$, *there exists* $A \in \mathbb{R}^{k \times n}, b \in \mathbb{R}^k, C \in \mathbb{R}^k, \beta \in \mathbb{R}$ *such that*

$$g(x) := C \cdot (\sigma \circ (Ax + b)) - \beta x$$
$$\sup_{x \in K} \|f(x) - g'(x)\| < \varepsilon$$

*Proof.* Given the mapping $f$, by UAT, there exists $A, b, C$ that approximate $f(x)$. Construct $\tilde{C} \in \mathbb{R}^k$ and $\beta \in \mathbb{R}$, such that

$$\tilde{C}_j = \begin{cases} C_j / A_{col,j}, & A_{col,j} \neq 0 \\ 0, & A_{col,j} = 0 \end{cases}, \quad \text{and} \quad \beta = \sum_{j | A_{col,j} = 0} C_j \sigma(b_j)$$

Then,

$$\sup_{x \in K} \|f(x) - C \cdot (\sigma \circ (A \cdot x + b))\|$$

$$= \sup_{x \in K} \left\| f(x) - \sum_{j=1}^{k} C_j (\sigma(A_j \cdot x + b_j)) \right\|$$

$$= \sup_{x \in K} \left\| f(x) - \sum_{j=1}^{k} \tilde{C}_j (\sigma(A_j \cdot x + b_j) A_{col,j}) - \sum_{j | A_{col,j} = 0} C_j \sigma(b_j) \right\|$$

$$= \sup_{x \in K} \|f(x) - \tilde{C} \cdot (\sigma' \circ (A \cdot x + b) \circ A_{col}) - \beta\|,$$

Note that $\tilde{C} \cdot (\sigma' \circ (A \cdot x + b) \circ A_{col}) - \beta = \dfrac{d}{dx_{col}} \left( \tilde{C} \cdot (\sigma \circ (Ax + b)) - \beta x \right)$, which is the derivative net of a two-layer feedforward integral network.

# F   Relationship between Number of ProdNets and Model Expressivity

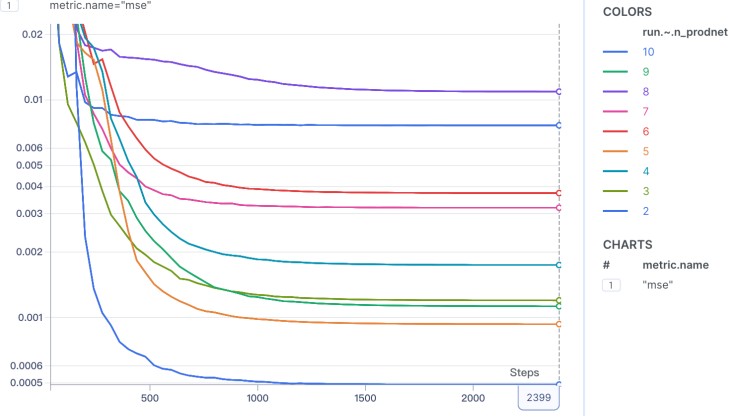

Figure 9: Training MSE for fitting a positive derivative network to $\sin(x) \cos(y) \sin(z) + 1$

We applied ten summations of positive ProdNets to fit the non-multiplicative-decomposable function $\sin(x) \cos(y) \sin(z) + 1$.

Each of these ProdNets consists of three MLP components, each with two hidden layers with 128 dimensions. All models underwent training with a consistent learning rate set at 0.005.

Our results, shown in Figure 9, indicate that increasing the number of ProdNets generally improves the model's performance in fitting the non-decomposable function. The model with the best MSE

uses 10 ProdNets, while the model with 2 ProdNets had the second-worst performance. This result is intuitively sensible, as more linear terms are typically required to express an arbitrary function precisely. However, we observed that employing more ProdNets does not always lead to better performance, as demonstrated by the model with 8 ProdNets.

