# OpenReview forum: "Automatic Integration for Spatiotemporal Neural Point Processes"
_NeurIPS.cc/2023/Conference — NeurIPS 2023 poster_

### Official Review · Reviewer_WN8G · 2023-06-30

**Soundness:** 3 good
**Presentation:** 3 good
**Contribution:** 3 good
**Rating:** 5
**Confidence:** 3

**Summary:**

In this paper, the authors propose a new method for spatiotemporal neural point processes (STPP). A key innovation of their method is that they introduced AutoInt for efficient integration of the intensity function of STPP. Using the divergence theorem and Green’s theorem, they convert the original triple integral to line integrals which AutoInt can efficiently calculate. They also impose 3D non-negativity constraint on the integral network. Experimental results show efficacy of the proposed method


**Strengths:**

1. Applying AutoInt to solve the STPP problem looks pretty novel.
2. Converting the original triple integral to line integrals using the divergence theorem and Green’s theorem and imposing non-negative constraint on the integral network also have strong methodology contribution.


**Weaknesses:**

1. The paper misses an important reference which also uses AutoInt for point processes: Automatic Integration for Fast and Interpretable Neural Point Processes. So the authors are encouraged to clarify contributions of this paper over the published paper above.

**Questions:**

1. It’s not clear why we cannot use AutoInt to do triple automatic integration on the intensity function directly.
2. Compared with traditional methods on integral calculation, e.g. Monte Carlo, what are the advantages of the proposed method from the viewpoint of computational complexity?
3. To train AutoInt (or corresponding duo neural networks), are there any minimum requirement of the number of events (training samples)?

**Limitations:**

Yes

---

> ### Author Rebuttal · Authors · 2023-08-09
>
> Thank you for your review of our paper. We appreciate your positive feedback on the novelty and methodology contribution of our proposed method.
>
> **Q: So the authors are encouraged to clarify contributions of this paper over the published paper above.**
>
> **A:** This paper was scheduled to appear in the 5th L4DC conference (June 14th~16th), but it has not been published or made publicly accessible before the NeurIPS submission deadline (May 11). Hence, we were unable to include it in our paper prior to submission. According to the NIPS [two month policy](https://neurips.cc/Conferences/2023/PaperInformation/NeurIPS-FAQ), authors are **not** expected to compare to work that appeared only a month or two before the deadline.
>
> Nevertheless, we recognize the significance of their work and will ensure to include it in the final version of our manuscript. We will also discuss the additional challenges our proposed method addresses compared to their problem settings.
>
> **Q: It’s not clear why we cannot use AutoInt to do triple automatic integration on the intensity function directly.**
>
> **A:** Figure 2 illustrates why direct extension of AutoInt for triple automatic integration is not feasible due to convexity challenges. If we directly constrain the mixed triple derivative (df / dxdydt) of a neural network to be positive by setting the whole network monotone, all triple derivatives would become positive. This would lead to an overly constrained model. For instance, df / dtdtdt is also positive, leading to an intensity that skyrockets on the time axis. As a result, the intensity function is approximated using a piecewise convex function, as shown in the third plot of Figure 2. To overcome this issue, we propose the ProdNet technique.
>
> **Q: Compared with traditional methods on integral calculation, e.g. Monte Carlo, what are the advantages of the proposed method from the viewpoint of computational complexity?**
>
> **A:** The main advantages include (1) sample complexity and (2) memory efficiency. The computational efficiency comparison with other integration methods heavily depends on the number of Monte Carlo samples. With a small number of samples (10 per dimension), Monte Carlo can be faster than our method but suffers from large numerical errors, as depicted in Figure 3. With a large number of samples, Monte Carlo has high memory usage, and backpropagation needs to be done sequentially for each batch of samples, making it inevitably slower than AutoInt.
>
> **Q: To train AutoInt (or corresponding duo neural networks), are there any minimum requirements for the number of events (training samples)?**
>
> **A:** There is no strict minimum requirement on the number of events (training samples) for training AutoInt networks. The training process can be performed with a small number (e.g. tens of) of event sequences, and one can always add a loss of regularity to avoid the overfitting of intensity function.

---

> > ### Comment · Reviewer_WN8G · 2023-08-18
> >
> > Thanks for the clarifications.

---

### Official Review · Reviewer_wohe · 2023-07-03

**Soundness:** 3 good
**Presentation:** 3 good
**Contribution:** 3 good
**Rating:** 6
**Confidence:** 4

**Summary:**

The paper proposes a novel framework to extend the automatic integration of conditional intensity function to spatio-temporal point processes, aiming to overcome the computational challenges of triple integration. Specifically, the proposed model adopts an integral network and converts the triple integral into line integrals using the Divergence theorem and Green’s theorem. The authors use a summation of ProdNet to handle the non-negativity of the conditional intensity function (derivative network) instead of directly imposing a non-negative constraint on the model parameters, which enhances the model expressiveness. The consistency, validity, and computational efficiency of the AutoInt STPP are comprehensively demonstrated.

**Strengths:**

1. The proposed framework based on auto-integration is valuable for spatio-temporal point process modeling. It significantly improves the model flexibility and accuracy compared to the close-form or monte-carlo based integration model.
2. The natural constraint of the spatio-temporal point processes (e.g., non-negativity of the conditional intensity function) is well and not overly satisfied. The triple integral has also been handled in a novel way.
3. The comprehensive experimental results on both synthetic and real-world data, including the ablation study, well support the claimed advantage of the proposed model.
4. The consistency of the AutoInt PP estimator is proved under mild constraints.

**Weaknesses:**

1. Although it does not impact the overall goodness of the paper, I have to admit that the presentation of Sec. 3.5 is not straightforward. The purpose and advantage of using line integral to handle the triple integral are not clear to me. It would be better if the authors could further explain it with several high-level sentences.
2. More experiments on the spatio-temporal Hawkes process with a non-separable influence function can be included. For example, the model performance on synthetic data generated by the ETAS [1], which is a widely-used STPP model in real-world practice.
3. I would say the performance advantage of AutoSTPP is not significant or highlighted. For example, it seems like the DSTPP has a similar performance to AutoSTPP on synthetic data, and so does MonteSTPP if the standard deviation is considered on real data.


---
[1] Space-Time Point-Process Models for Earthquake Occurrences. Yosihiko Ogata. Annals of the Institute of Statistical Mathematics, 1998, vol. 50, issue 2, 379-402

**Questions:**

1. Forgive me if I understand wrong, but is there an integral of f over S missed in the integral at line 151?
2. I would like to know how N (in ProdNet) is chosen in experiments, and if there is a significant performance difference with different N.
3. It would be great if the authors could explain the advantages of AutoSTPP over DSTPP.
4. The computational efficiency of the derivative is great, however, the readers or practitioners would like to know the comparison of the computational efficiency of the model with other baselines.

**Limitations:**

1. Modeling performance and computational efficiency can be further explained or demonstrated.

---

> ### Author Rebuttal · Authors · 2023-08-09
>
> Thank you for your review of our paper. We appreciate your positive feedback on the proposed framework and the comprehensive experimental results supporting our claims. We acknowledge the weaknesses and limitations pointed out and will address them accordingly.
>
> **Q: I have to admit that the presentation of Sec. 3.5 is not straightforward. The purpose and advantage of using line integral to handle the triple integral are not clear to me.**
>
> **A:** Regarding the presentation of Section 3.5, we apologize if it was not clear enough. The triple integral is necessary for computing the likelihood during the training stage. The line integral is not particular; we convert the triple integral over a cuboid to double integrals over 6 squares, then line integrals over 24 lines. It is a consequence of sequential application of Divergence theorem, Green’s theorem and the Fundamental theorem of calculus. AutoInt guarantees that the triple, double and line integral all have closed forms. We will make sure to provide a clearer high-level explanation in the updated draft.
>
> **Q: More experiments on the spatio-temporal Hawkes process with a non-separable influence function can be included. For example, the model performance on synthetic data generated by the ETAS [1].**
>
> **A:** We appreciate the suggestion. However, the intensity of the ETAS model is still additively separable, based on the equation mentioned in the 4th page of [1]. The current synthetic spatio-temporal Hawkes process in our paper is a simplified version of ETAS without the term t^{1+\beta}. Indeed, adding more point processes with intensity that is not additively separable would be valuable. That is why we included the spatiotemporal extension of the self-correcting process in the synthetic experiments.
>
> **Q: I would say the performance advantage of AutoSTPP is not significant or highlighted.**
>
> **A:** We agree that the advantages of AutoSTPP over DSTPP should be discussed more. One major advantage of AutoSTPP is its ability to learn any intensity without relying on kernel density estimation. This not only reduces the number of hyperparameters but also improves flexibility. When examining the estimated intensity of the self-correcting process, one may notice that DSTPP incorrectly approximates the underlying intensity with a mixture of Gaussian kernels, while AutoSTPP learns a much smoother intensity. We will include a more detailed discussion in the final draft.
>
> **Q: Forgive me if I understand wrong, but is there an integral of f over S missed in the integral at line 151?**
>
> **A:** It's a typo and thank you for pointing it out!
>
> **Q: I would like to know how N (in ProdNet) is chosen in experiments, and if there is a significant performance difference with different N.**
>
> **A:** Please refer to Appendix F: while N=2 could reduce the performance by a margin, further increasing the number of ProdNet has limited effect on performance, and does not always boost the performance. It shows that N=5 can already cover a sufficient number of functions. We selected N by this gridsearch and fixed it to be 10 during experiments.
>
> **Q: The computational efficiency of the derivative is great, however, the readers or practitioners would like to know the comparison of the computational efficiency of the model with other baselines.**
>
> **A:** Our current records show that both DSTPP (\~1 hour) and AutoSTPP (1\~3 hour) are much faster than the normalizing-flow-based NSTPP (> 12 hours). AutoSTPP is often slower than DSTPP due to the larger computation graph, which can be influenced by the choice of parameter N. We will include a table containing the multirun averages in the appendix to provide a comprehensive comparison.

---

> > ### Comment · Reviewer_wohe · 2023-08-18
> > **Response to Authors**
> >
> > I appreciate the detailed response to my questions from the authors, while there are plenty of aspects for the paper to be improved. I intend to keep the current score.

---

### Official Review · Reviewer_Hisf · 2023-07-08

**Soundness:** 3 good
**Presentation:** 3 good
**Contribution:** 3 good
**Rating:** 6
**Confidence:** 4

**Summary:**

The paper introduces a new approach to address the challenge of integrating spatiotemporal point processes in a flexible and efficient manner. The existing methods either lack flexibility or introduce numerical errors. The authors propose AutoSTPP as a novel paradigm that overcomes this limitation and showcases its advantages in recovering complex intensity functions from irregular spatiotemporal events. Its effectiveness is validated through experiments on synthetic data and benchmark real-world datasets.

**Strengths:**

1. Clear presentation. The writing is clear and concise.
2. It provides the proof of consistency for AutoSTPP, which enhances the credibility of the proposed method.
3. Comprehensive experiments. The authors evaluate the proposed approach on both synthetic datasets and real-world datasets.

**Weaknesses:**

1. An essential related work is missing: Automatic Integration for Fast and Interpretable Neural Point Processes [1].
2. Despite the solution for STPP is very interesting, the proposed technique lacks novelty. Zhou et al.[1] has already proposed a very similar framework for TPPs. The major different part lies in the data dimension. I don’t think just extending 1D to 3D is qualified for an accepted NIPS paper.

[1] Zhou, Zihao, and Rose Yu. "Automatic Integration for Fast and Interpretable Neural Point Processes." Learning for Dynamics and Control Conference. PMLR, 2023.

## After Rebuttal

The authors' response has addressed most of my concerns. Therefore, I decide to raise the score to 6 (Weak accept).

However, I highly suggest authors carefully discuss links between their work and Zhou et al. [1].

Moreover, recently developed diffusion (or score-based) models seem like a promising workaround for handling likelihood computation in the neural point process. Authors should discuss the following related works that have been published recently.
- Li, Zichong, et al. "SMURF-THP: Score Matching-based UnceRtainty quantiFication for Transformer Hawkes Process." in ICML 2023.
- Yuan, Yuan, et al. "Spatio-temporal Diffusion Point Processes." in KDD 2023.

**Questions:**

What’s the key contribution of this paper compared to [Zhou et al 2023]?  The authors should carefully differentiate their approach from it.

---

> ### Author Rebuttal · Authors · 2023-08-09
>
> Thank you for providing feedback on our paper. We appreciate your positive remarks regarding the soundness and presentation of the paper. Below, we address the concerns you raised regarding the contribution.
>
> **Q: An essential related work is missing: Automatic Integration for Fast and Interpretable Neural Point Processes [1].**
>
> **A:** This paper was scheduled to appear in the 5th L4DC conference (June 14th~16th), but it has not been published or made publicly accessible before the NeurIPS submission deadline (May 11). Hence, we were unable to include it in the related work in our paper prior to submission. According to the NeurIPS [two month policy](https://neurips.cc/Conferences/2023/PaperInformation/NeurIPS-FAQ), Papers appearing less than two months before the submission deadline are generally considered concurrent to NeurIPS submissions. authors are **not** expected to compare to work that appeared only a month or two before the deadline, let alone **after the deadline**.
>
> Nevertheless, we recognize the significance of their work and will ensure to include it in the final version of our manuscript. We will also discuss the additional challenges our proposed method addresses compared to their problem settings.
>
> **Q: The major different part lies in the data dimension. I don’t think just extending 1D to 3D is qualified for an accepted NIPS paper.**
>
> **A:** We respectfully disagree. Despite the naming similarity of AutoInt between our approach and the work by [Zhou et al.], the extension from 1D to nD (n>1) is a highly **non-trivial contribution**. Zhou et al. can only learn temporal point processes without markers, whereas our paper focuses on extensible spatiotemporal point processes.  The difference in data dimension makes the problem significantly harder, hence introducing several technical differences. One of the major challenges is the inherent difficulty of constraining the sign of higher-order derivatives without compromising the performance. As detailed in Section 3, if we simply extend the 1D AutoInt technique in [Zhou et al], we would overconstrain the model and fail to recover the ground truth intensity. Another technical challenge is the computational complexity. The computational cost of AutoInt scales exponentially with the number of dimensions. Our proposed method tackles these technical challenges through many novel techniques including ProdNet and Divergence theorem. We circumvent the trivariate computational graph and use simplified univariate graphs to make AutoInt scales with the number of dimensions.

---

> > ### Comment · Reviewer_Hisf · 2023-08-21
> > **Thanks for your response**
> >
> > Thanks for your response. It has addressed most of my concerns and I have raised my score to 6. My remaining concern on discussing more related works has been updated in ``Weaknesses''.

---

### Official Review · Reviewer_HnTF · 2023-07-27

**Soundness:** 4 excellent
**Presentation:** 4 excellent
**Contribution:** 3 good
**Rating:** 7
**Confidence:** 4

**Summary:**

In this paper is proposed an automatic integration scheme for efficient learning for spatiotemporal process models, where the corresponding function is approximated with a deep neural network. A priori, the method can be used to compute the likelihood of any intensity, even for sophisticated ones. The method is validated in both synthetic and real datasets, where the proposed method obtains better results than those provided by state-of-the-art approaches.

**Strengths:**

The paper is well written and clear enough. In general, the authors provide most of the details, explanations and assumptions to understand their approach and the corresponding contribution. To be honest, I do not have many issues with the current submission.

The method shows a clear advance in terms of accuracy in comparison with imposing the constraint through activation function with a nonnegative triple derivative. AutoSTPP performance is very competitive with respect to previous neural STPPs.

Both quantitative and qualitative experiments are provided in the paper. Both of them show the superiority of the method, validating all the claims in the paper. In particular, I like the experiments on synthetic data, as complex intensity functions are considered and the proposed method can capture them correctly. In any case, the real-world scenarios are also needed, and the set of experiments in this line is well executed. Moreover, the method clearly outperforms the rest of competing methods in these experiments.

**Weaknesses:**

Some parameters in Eq.(1) are not properly defined. See, u or \tau, for example. I know many parameters are included in the paper so please, be careful with all of them.

The authors do not include a proper discussion about the limitations of the method as it is a bit superficial. Moreover, some failure cases could help the reader. To this end, including more complex intensity functions where the method obtains a worse performance could greatly improve the paper. Thanks to that, we can see the real limitations of the method.

I am not sure why the authors only use two real-world datasets of Chen et al. 2020, as more cases were considered there. At least, that could be briefly commented.

**Questions:**

As the proposed method is available for learning continuous-time dynamics, I think the authors could consider more challenging scenarios where the applicability is especially relevant in science and technology, such as fluids or continuous mechanics. I would like to see at least a discussion about this type of scenario.

Ablation studies by considering the parameters in table 3. The authors should indicate how the best parameters were selected, and if any variation is given in the parameter tuning for both synthetic and real experiments.

**Limitations:**

The authors do not include a proper discussion about the limitations of the method as it is a bit superficial. Moreover, some failure cases could help the reader. To this end, including more complex intensity functions where the method obtains a worse performance could greatly improve the paper. Thanks to that, we can see the real limitations of the method.

---

> ### Author Rebuttal · Authors · 2023-08-09
>
> Thank you very much for reviewing our paper. We greatly appreciate the reviewer's thorough feedback and the positive evaluation of our paper.
>
> **Q: Some parameters in Eq.(1) are not properly defined.**
>
> **A:** Sorry for not properly defining tau in Equation (1). Thank you for pointing this out! We will make sure to address this in the updated manuscript.
>
> **Q: The authors do not include a proper discussion about the limitations of the method as it is a bit superficial.**
>
> **A:** We agree that a more comprehensive discussion on the limitations and potential failure cases of our method would be valuable. One limitation is the "same influence" assumption, which may introduce an inductive bias when the influence of different events significantly differs. However, our empirical findings suggest that removing this assumption can lead to worse performance, presumably due to a lack of estimator consistency. Exploring this limitation further would be an interesting avenue for future research.
>
> **Q: I think the authors could consider more challenging scenarios where the applicability is especially relevant in science and technology, such as fluids or continuous mechanics.**
>
> **A:** Thank you for your suggestion. Fluids or continuous mechanics are not discrete event data, hence are not directly applicable to our paper. The earthquake datasets we used in the paper are challenging datasets for event prediction.
>
> We stress that the selection of datasets in our experiments do some favors for the baselines (in fact, they are from the baseline papers) models and are relatively challenging for our model. Our synthetic datasets, such as the spatiotemporal Hawkes dataset, feature Gaussian influence functions that are assumed by DSTPP. Incorporating a dataset with more complex influence functions, like a sine wave, while still maintaining the "same influence" assumption, could provide a clearer performance gap compared to baseline methods.
>
> **Q: Ablation studies by considering the parameters in Table 3.**
>
> **A:** We follow the simulation parameter settings in the DSTPP paper, and they are only for generating the datasets. One advantage of our proposed method is that it has very few hyperparameters (number of ProdNet and hyperparameters for MLP).

---

### Author Rebuttal · Authors · 2023-08-09

Once again, we sincerely thank you for your valuable feedback and constructive criticism. We will incorporate the suggested improvements and address the raised concerns in the final version of the manuscript. We will emphasize the differences between our approach and the work by Zhou et al. published after the submission.

---

> ### Author Response · Authors · 2023-08-10
>
> We express our gratitude to the reviewers for their considerate feedback.
>
> We appreciate that they found our writing to be clear (Hisf, HnTF) and our AutoInt idea to be both interesting and innovative (Hisf, whoe, WN8G). It is reassuring to know that they acknowledge our approach's advantages over baselines, as supported by comprehensive experiments conducted on both synthetic and real-world datasets (Hisf, HnTF, whoe). We are particularly pleased that HnTF recognizes the importance of validating our approach through visualizing the recovery of the ground truth intensity function.
>
> Furthermore, we would like to extend our gratitude to Hisf and WN8G for bringing to our attention the recent paper by Zhou et. al, which employs a similar approach but does not involve ProdNet and Divergence theorem, two significant technical contributions of our paper. Unfortunately, as the paper was not publicly accessible before the NeurIPS submission deadline, we are unable to provide a direct comparison. While we are not expected to compare with it in accordance with the NeurIPS two-month policy, we acknowledge its significance and intend to include it in the final version of our manuscript.
>
> Additionally, we would like to highlight the considerable challenge that extending the approach from 1D to nD (where n > 1) represents, notwithstanding the work done by Zhou et. al. Their approach only applies to learning temporal point processes without markers, while our paper focuses on extensible spatiotemporal point processes. The extension to nD introduces numerous additional technical hurdles, such as ensuring model flexibility and managing computational complexity. As explained in detail in Section 3, if we were to simply extend the 1D AutoInt proposed by Zhou et al., we would overconstrain the model and fail to recover the ground truth. Moreover, the computational cost of AutoInt scales exponentially with the number of dimensions. To address these challenges, our proposed method effectively decomposes the joint probability into marginal probabilities using ProdNet. By leveraging simplified univariate graphs, we are able to circumvent the computational complexities associated with the trivariate computational graph.

---

### Decision · Program_Chairs · 2023-09-21

**Decision:**

Accept (poster)

**Comment:**

The manuscript proposes an automatic integration scheme for efficient learning for spatiotemporal process models. While the initial reviewers were mixed, the authors quite satisfactorily answered the questions raised by the reviewers. This is also reflected in the final scores of this manuscript. I also stand with the authors on the NeurIPS *two-month policy*. However, it would be greatly appreciated to include a discussion on the differences and links to the work of Zhou et al. in the final version. Considering all the initial reviews, the rebuttal, and the reviewer discussion, I do recommend "Accept".